# A situational assessment of treatments received for childhood diarrhea in the Federal Republic of Nigeria

Ezra Gayawan[1,2]☯*, Ewan Cameron[2,3]☯, Tolu Okitika[2], Osafu Augustine Egbon[1,4], Peter Gething[2,3]

**1** Department of Statistics, Federal University of Technology, Akure, Nigeria, **2** Geospatial Health & Development, Telethon Kids Institute, Nedlands, Western Australia, Australia, **3** School of Population Health, Curtin University, Bentley, Western Australia, Australia, **4** Department of Statistics, Federal University of São Carlos, São Carlos, Brazil

☯ These authors contributed equally to this work.
* egayawan@futa.edu.ng

**Data Availability Statement:** The data underlying the results presented in the study are available from https://www.kaggle.com/datasets/ezragayawan/diarrhea-treatment-data.

## Abstract

We assess progress towards improved case management of childhood diarrhea in Nigeria over a period of targeted health systems reform from 2013 to 2018. Individual and community data from three Demographic and Health Survey rounds are leveraged in a geospatial model designed for stratified estimation by venue of treatment seeking and State. Our analysis reveals a highly regionalised health system undergoing rapid change. Nationally, there have been substantial increases in the proportion of children under 5 years old with diarrhea receiving the recommended oral rehydration therapy after seeking treatment at either a health clinic (0.57 [0.44–0.69; 95% CI] in 2008; 0.70 [0.54–0.83] in 2018) or chemist/pharmacy (0.28 [0.17–0.42] in 2008; 0.48 [0.31–0.64] in 2018). Yet State-level variations in venue attendance and performance by venue have conspired to hold the overall proportion receiving this potentially life-saving therapy (0.45 [0.35–0.55] in 2018) to well-below ideal coverage levels. High performing states that have demonstrated significant improvements include Kano, Jigawa and Borno, while under-performing states that have suffered declines in coverage include Kaduna and Taraba. The use of antibiotics is not recommended for mild cases of childhood diarrhea yet remains concerningly high nationally (0.27 [0.19–0.36] in 2018) with negligible variation between venues. Antibiotic use rates are particularly high in Enugu, Kaduna, Taraba, Kano, Niger and Kebbi, yet welcome reductions were identified in Jigawa, Adamawa and Osun. These results support the conclusions of previous studies and build the strength of evidence that urgent action is needed throughout the multi-tiered health system to improve the quality and equity of care for common childhood illnesses in Nigeria.

## Introduction

Diarrheal illnesses are amongst the leading causes of childhood mortality in Nigeria, accounting for 16% of child deaths [1]. Oral rehydration solution (ORS), a mixture of clean water, salt

**Funding:** Gates Foundation [INV-009390 OPP1197730] Peter Gething was additionally supported through funding provided jointly by Curtin University, the Telethon Trust and the Telethon Kids Institute under project ID RES-61992.

**Competing interests:** The authors have declared that no competing interests exist.

and sugar, and preferably administered with zinc supplementation, has been shown to reduce the mean duration of diarrhea (by about 20%), mortality due to diarrhea (by 23%), stool output and the risk of a subsequent episode in the 2 to 3 months after supplementation [2–4]. However, usage of ORS is reported to be low in most low- and middle-income countries [5, 6]. The use of recommended home fluid (RHF) was promoted by the World Health Organization (WHO) in addition to ORS and the combination is referred to as oral rehydration therapy (ORT). As a non-packaged home fluid alternative, RHF can comprise measured sugar and salt added to clean water, or other home fluids such as rice water, coconut water, juice or tea [7, 8].

Sustainable Development Goal (SDG) 3.8 is to achieve universal health coverage, including access to essential health care services and access to safe, effective, quality and affordable essential medicines for all. However, there has been slow progress in increasing access to simple and affordable treatments in Nigeria. The proportion of children with diarrhea who received either ORS or RHF increased from 31% to 42% over a ten-year period (2008-2018), while those for whom treatment was sought rose from 42% to 65% [9]. These national figures, however, mask substantial local variations in common with other health indicators in Nigeria [10, 11]. About 3% of the children from Taraba state that had diarrhea in 2018 received ORS and zinc supplements compared with 72% in Nasarawa state [5]. Appropriate and timely care-seeking in formal health facilities would allow for prompt and correct diagnosis, appropriate management and avert complications [12, 13].

Many Nigerian mothers have adequate knowledge of what constitutes diarrhea and could identify episodes of it in their children [14]. But poor knowledge of the use of ORS and zinc supplementation has been identified as a major barrier to improving usage across different parts of Nigeria [1]. There are mothers who hold the myth that diarrhea is a sign that the child is teething, which can negatively influence their health care decision making [1, 15]. A study in eight Nigerian States, and another in Oyo State, identified unavailability, unaffordability and poor awareness as the major barriers to ORS/zinc utilization [13, 16]. Encouragingly, consultation with a health worker, having a secondary or higher level of education and exposure to mass media were reported to encourage the use of ORT for the treatment of diarrhea [17]. The venue where treatment was sought also substantially affects the chances of receiving ORS or ORS and zinc supplementation across the country [13].

Because it is rarely caused by bacteria, the use of antibiotics for infectious diarrhea brings little benefit in most cases [18]. The WHO guidelines on the treatment of diarrhea therefore discourages the use of antibiotics for treating non-severe cases, restricting its use to instances of bloody diarrhea and cholera with severe dehydration [19, 20]. Indiscriminate use of antibiotics leads to antibiotic resistance, capable of compromising the treatment of infectious diseases and undermining other advances in health and medicine [19]. Over-use of antibiotics also leads to increased medical costs, longer hospital stays and increases the risk of subsequent diarrhea episodes because of its effects on gut microflora [21]. Notwithstanding, studies have reported abundant use of antibiotics in treating mild cases of diarrhea in various parts of Nigeria [20–23].

Over the years, intervention strategies have been put in place in Nigeria to enhance easy access to life-saving treatments for childhood illnesses. Prominent here is the Essential Childhood Medicines Scale-up Plan (ECMSP) (2012-2015) that was contemplated as the country's first national road map for increasing access to life-saving treatments [24]. Aligning with the proposed strategies and activities under the ECMPS, the Clinton Health Access Initiative (CHAI) carried out some program activities in eight Nigerian states between 2013 and 2017 that aimed to address demand, supply and policy barriers to ORS and zinc uptake in both public and private sectors [13]. We set out to quantify trends by venue in treatment seeking behavior for childhood diarrhea among under-five children according to States of Nigeria. Our

objective is to quantify progress in diarrhea treatment in the country over a ten-year period considering the various interventions and programs put in place over the years. We used data from three waves of Nigeria Demographic and Health Surveys (NDHS) conducted in 2008, 2013 and 2018 and adopted a Bayesian geostatistical modeling technique that incorporates the sampling characteristics of the surveys. This allows for more accurate estimates of the treatment seeking behavior at State levels and provides actionable insight for policy makers at and above this adminstrative scale.

## Materials and methods

### Datasets

Dataset was compiled from three waves of the NDHS conducted in 2008, 2013 and 2018. Implemented by the Nigeria Population Commission with technical support from the DHS Program, these nationally-representative surveys offer quality assured cross-sectional data on a diverse collection of demographic and health indicators. The survey team used standardised methodology that facilitates comparative analyses across time and space. The team used a two-stage stratified cluster design for data collection. First, enumeration areas (clusters) were selected from a sampling frame created for the 2006 Population and Housing Census of the Federal Republic of Nigeria; and second, households were selected from each cluster by equal probability sampling. They randomly selected a total of 777, 904 and 1,400 clusters for the 2008, 2013 and 2018 surveys, respectively, from which representative samples of 34,070, 40,680 and 42,000 households were constructed. The response rates for the eligible women in each survey, who must be between ages 15 and 49 years, were 97%, 98% and 99%.

The survey team collected information on child health and interactions with the local health system for children under 5 y/o living in the surveyed households. They collected data concerning experiences of ill health using questionnaire targeting events in the two weeks prior to the surveys, answered on the child's behalf by the mother or caregiver. A child was identified to have suffered from diarrhea if the child had three or more loose or liquid stools in the same day (on one or more days of the preceding two weeks). Whenever a child is positively identified as having experienced diarrhea, the team asked the caregiver if the child was given oral rehydration/ORS or antibiotic pills or syrups. If the child received any treatment, the caregiver was asked to indicate where the treatment was sought. The response variables for our analysis were constructed as binary indicators as for: (i) each of ORS/RHF and antibiotic, indicating whether or not the child was treated with any of these and (ii) each of treatment-seeking class indicating whether or not treatment was sought at a health facility (private or public facility), a chemist/pharmacy or treated at home.

Nigeria, an African country located in the Golf of Guinea, is a multinational State with over 200 million inhabitants of more than 250 ethnic groups, all identifying with diverse cultural belief. Administratively, the country is divided into six geopolitical zones with each comprising of about six States and the Federal Capital Territory, Abuja as indicated in Fig 1.

### Geostatistical model

Given the multistage structure employed in gathering the DHS data, a key element in geostatistical modeling is to include the sampling characteristics in the model, as doing otherwise could potentially bias the estimates. We adopt a geostatistical technique for areal data that considers the stratified multistage cluster design used in data collection as proposed by Chen and colleagues [25]. The approach proceeds by first computing the proportion for each location

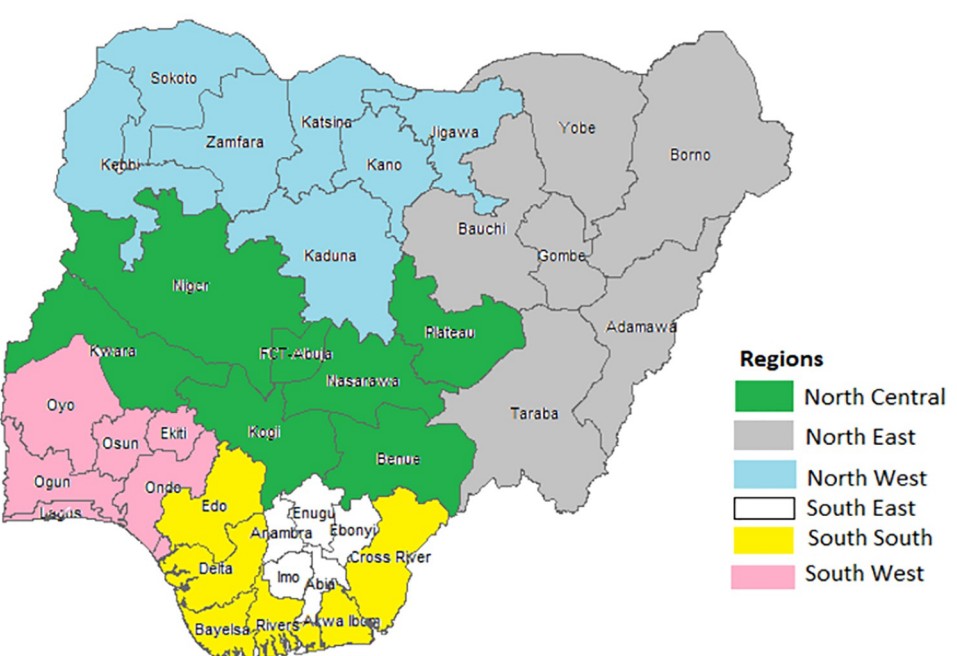

**Fig 1. Map of Nigeria showing the thirty six States and the FCT, Abuja grouped according to their geopolitical zones.**

using a designed-based weighted (direct) estimator:

$$p_i^{HT} = \frac{\sum_{c \in s_i} \sum_{k \in s_c} w_{ck} Y_{ck}}{\sum_{c \in s_i} \sum_{k \in s_c} w_{ck}} \tag{1}$$

where $c = 1 \cdots C$ are clusters sampled within strata, and for each cluster $c$, $k \in s_c$ is the indexes of individuals sampled in each cluster $c$. The binary outcome $Y_{ck}$ is the quantity whose prevalence is to be estimated, $w_{ck}$ are the sampling weights and $s_i$ is the indices of clusters sampled in area $i$. To enhance precision, a transformation of this weighted estimate can thereafter be modeled using a random effect model [26]. An approach for the binary outcome is to consider $Z_i = \text{logit}(p_i^{HT})$. If the design-based variance is denoted by $V_i$, the model is

$$Z_i \sim N(\mu_i, V_i)$$

$$\mu_i = S_i + \epsilon_i$$

where $\epsilon_i \sim_{iid} N(0, \sigma_\epsilon^2)$ for $i = 1 \cdots n$ and $S \in [1 \cdots 37]$ are the spatial locations (States of Nigeria) that are assigned a spatial distribution. Thus, an intrinsic conditional autoregressive (ICAR) prior, following the assumption that in general, events at neighboring locations are likely to be similar, was adopted. We implement the model using the SUMMER package in R. The package is useful for small area estimation using survey data collected with complex stratification designs [27]. The package builds on the R-INLA package using the BYM parameterisation for estimating the smoothed spatial effects [28]. We chose 90% credible intervals to quantify the significance of our estimates. To compare changes in events between the previous years and 2018 for each State, we subtracted the estimated prevalence between the years compared, while to determine the significance in the change, we estimated the difference in the credible intervals such that whenever (a) the upper 90% credible interval value for earlier year is greater

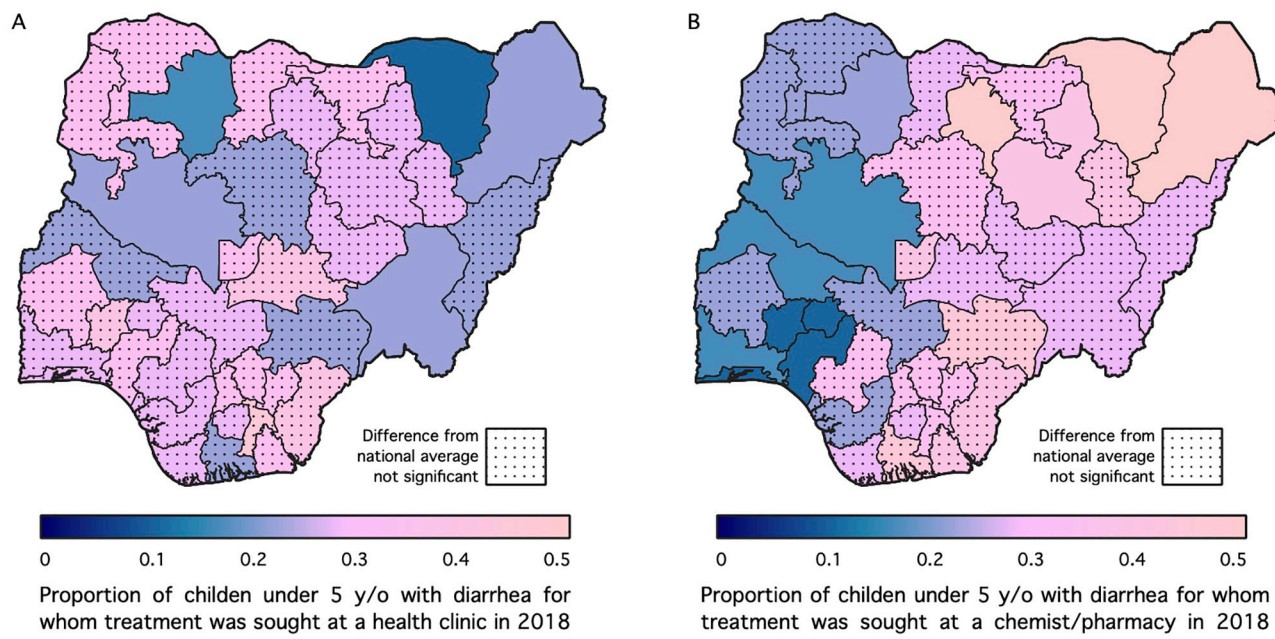

**Fig 2. Model-based estimates of the breakdown of treatment seeking for early childhood diarrhea in Nigeria by venue and state for 2018.**

than the lower 90% credible interval value for 2018 or (b) the lower 90% for the earlier year is lower than the upper 90% for 2018, the difference is said to be not statistically significant.

## Results

### Treatment seeking venues

Fig 2 presents the results for treatment seeking-behavior for diarrhea, indicating whether treatment was sought at a clinic (Fig 2A) or at a chemist/pharmacy (Fig 2B). For all the maps, States shaded with dots are places where the estimates are not significantly different from the national average. The findings reveal that about 20% or fewer of the children would be taken for treatment at a health facility among those who reside in Borno, Yobe, Taraba, Niger and Zamfara states while the estimates are not significant for the other states. In the case of chemist/pharmacy, a larger proportion of the children who live in Borno, Yobe, Bauchi and Kano states sought treatment for diarrhea at a chemist/pharmacy.

### Oral rehydration therapy

We estimated the national prevalence for the use of ORS/RHF received from any location to be 0.41 (90% CI: 0.31, 0.51) in 2008; 0.43 (90% CI: 0.32, 0.53) in 2013; and 0.45 (90% CI: 0.35, 0.55) in 2018, indicating a marginal increase of 4% over a ten-year period (Table 1). When broken down by venue, findings from the 2018 data show that the recommended therapy was received by 24% (90% CI: 11%, 35%), 48% (90% CI: 31%, 64%) and 70% (90% CI: 54%, 83%) of children for whom treatment was sought at home, a chemist/pharmacy or at a health facility, respectively Table 1. The stratification by venue offers potentials for interventions through the venues, however, given that our data source provides information on venue choice amongst those who received ORS, this could confound venue preferencing with performance.

**Table 1. National prevalence for the use of ORS/RHF and antibiotic based onlocation where it was received.** Values in parenthesis are the 90% credible intervals.

| | Venue | | | |
|---|---|---|---|---|
| Year | Home | Chemist | Clinic | Anywhere |
| | ORS/RHS | | | |
| 2008 | 0.23 (0.12, 0.36) | 0.28 (0.17, 0.42) | 0.57 (0.44, 0.69) | 0.41 (0.31, 0.51) |
| 2013 | 0.22 (0.12, 0.35) | 0.40 (0.26, 0.55) | 0.67 (0.52, 0.80) | 0.43 (0.32, 0.53) |
| 2018 | 0.24 (0.11, 0.35) | 0.48 (0.31, 0.64) | 0.70 (0.54, 0.83) | 0.45 (0.35, 0.55) |
| | Antibiotics | | | |
| 2008 | 0.16 (0.08, 0.27) | 0.48 (0.27, 0.43) | 0.45 (0.31, 0.59) | 0.32 (0.22, 0.42) |
| 2013 | 0.22 (0.11, 0.35) | 0.49 (0.33, 0.65) | 0.53 (0.38, 0.67) | 0.43 (0.31, 0.52) |
| 2018 | 0.13 (0.06, 0.24) | 0.35 (0.21, 0.53) | 0.35 (0.21, 0.51) | 0.27 (0.19, 0.36) |

The State-level results presented in Fig 3 indicate that the proportion of children with diarrhea who received the recommended ORS/RHF in Kaduna, Plateau, Taraba, Gombe, Bauchi and Yobe States are lower than the national average (about 30% or lower) but higher among those residing in Borno, Jigawa, Kano and Nasarawa States; estimates for all the other States, including the FCT, are not significant. In Fig 3B, we compute change in the proportion of children with diarrhea who received the recommended ORS/RHS therapy since 2008, with the aim of measuring State-specific progress over the ten-year period. The findings indicate significant positive change in Sokoto, Katsina, Kano, Jigawa, Bauchi, Borno and Nasarawa States but negative in Kaduna and Taraba, implying reduction in the proportion of children with diarrhea who received the recommended therapy in these two areas.

Furthermore, we compute the differences in proportion of children who received recommended ORS/RHF based on the venue where treatment was sought, considering each venue against all other venues (Fig 3C–3E). The estimates suggest that the use of ORS/RHS for treatment of diarrhea is modified by the place where treatment was sought. Specifically, with the exception of a few states namely, Borno, Nasarawa, Plateau, Ogun, Delta, Akwa Ibom and Rivers, where the estimates are not significant, the majority of the children from the other states for whom treatment was sought at a health facility would receive the recommended ORS/RHF. For those who sought treatment at a chemist/pharmacy, only Niger and Jigawa states show positive changes in the proportion of children who received the recommended therapy. Regarding the children for whom neither a health facility nor chemist/pharmacy was visited, the findings indicate negative change in the chances of receiving the recommended ORS/RHF in majority of the States, with no State having higher proportion in this category. Consequently, improvement in the use of ORS/RHF for treating diarrhea among children in Nigeria can be driven by care seeking at health facilities, similar to what was reported by Lam and colleagues [13].

## Antibiotic use

The national mean proportion of early childhood diarrhea cases treated with antibiotics, as shown in Table 1, was estimated to be 0.32 (90% CI: 0.22, 0.42) in 2008, 0.42 (90% CI: 0.31, 0.52) in 2013 and 0.27 (90% CI: 0.19, 0.36) in 2018. Venue specific estimates for 2018 were 0.13 (90% CI: 0.06, 0.24) for home, 0.35 (90% CI: 0.25, 0.53) for chemist/pharmacy and 0.35 (90% CI: 0.21, 0.51) for health sector. The State-specific estimates presented in Fig 4 show that a high proportion of the children who suffered from diarrhea and lived in Kaduna, Anambra, Niger, Taraba, Kano and Yobe (at least 70% in Kaduna and Anambra) would be treated with

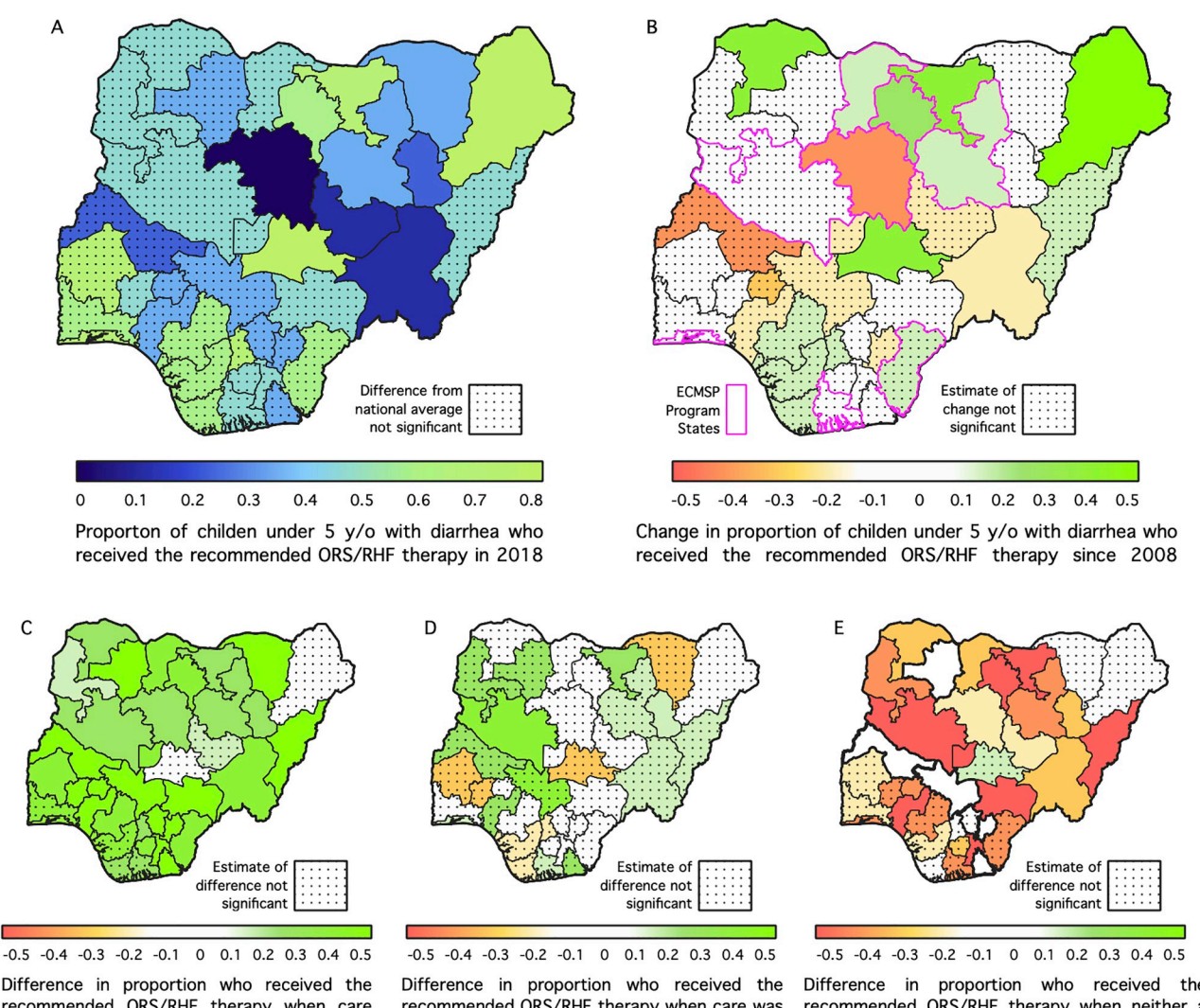

**Fig 3. Model-based estimates of the proportion of early childhood diarrhea cases receiving the recommended oral rehydration therapy in Nigeria by venue and state in 2018.**

antibiotics. A study had reported that about 85% of children with acute diarrhea were prescribed antibiotics at healthcare facilities in Abakaliki, South East, Nigeria [20] while in another study, about half of the children used unprescribed antibiotics in Enugu [22]. These point to the widespread use of antibiotics for acute diarrhea in the country. The estimates are also significantly high but to a lesser extent for those in Adamawa, Plateau, Jigawa, Katsina, Kogi, Ondo and Osun states. On the change in use of antibiotics since 2008, the results show negative changes in Adamawa, Jigawa and Osun states but positive in Taraba. For the differences in the proportion of children who received antibiotics when care was sought at a health facility against other venues, the results indicate that children who reside in Bauchi, Plateau, Kano, Jigawa, Adamawa, Zamfara and the FCT are more likely to receive antibiotics when treated at a health facility; those who lived in Kebbi, Sokoto, Kwara, Kaduna, Kano, Gombe, Yobe, Benue, Imo and Abia States and visited a chemist/pharmacy would more likely to have

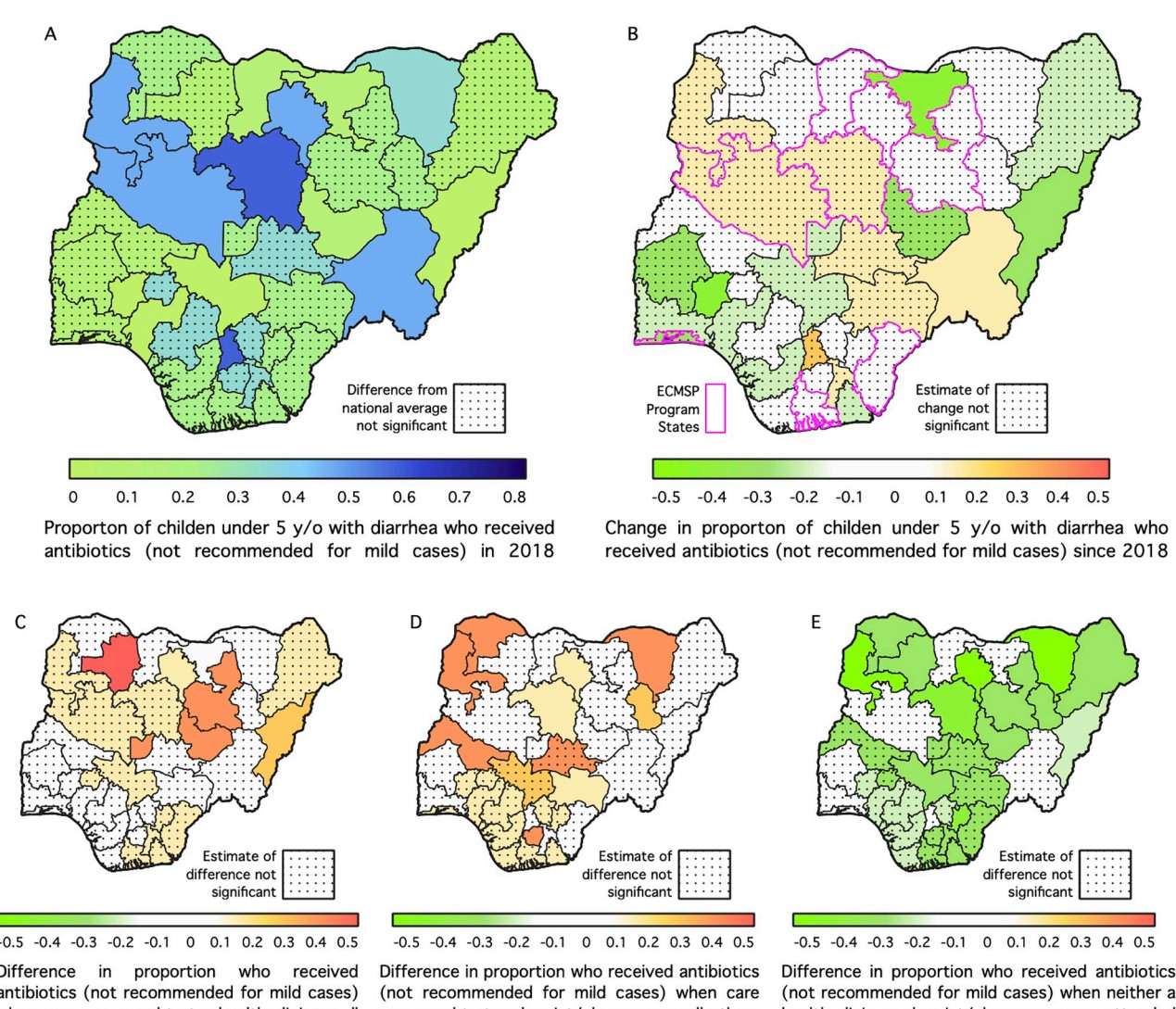

**Fig 4. Model-based estimates of the proportion of early childhood diarrhea cases receiving antibiotics in Nigeria by venue and state in 2018.**

been given antibiotics. In the majority of the States, those who neither visited a health facility nor a chemist/pharmacy were less likely to have been treated with antibiotics.

## Discussion

The treatment of childhood diarrhea in Nigeria has often been marred by poor knowledge of the use of appropriate low cost therapy [1]. A consequence is the increase in the number of episodes that young children incur in a year. There is a widespread use of antibiotics without diagnostic or epidemiological reasons to justify the use [20, 22]. Over the years, different interventions by local and international authorities to improve life-saving health services particularly to the most vulnerable population have been contemplated but these have yielded mixed impact at different communities [13]. Thus, model-based estimates empower investigations at subnational levels, improving the understanding of local variations and trends and aiding

targeted interventions. Geostatistical modeling endeavors in the Nigerian public health domain have therefore been of interest in recent years [10, 11].

This findings demonstrate sub-optimal use of formal healthcare services for child health throughout the country and particularly among individuals in the northern fringes as revealed by previous studies [12, 29, 30]. Some authors attributed this to the prevailing cultural differences in perception of health and diseases, and the availability and distance to healthcare facilities [30]. Facing health service inacessibility and/or unaffordability many individuals may turn to coping strategies such as the use of traditional medicine, patronizing unaccredited chemists or visiting prayer houses [31]. A systematic review reveals that in most low- and middle-income countries (LMIC) the majority of caregivers seek care for childhood illnesses but only a handful utilize formal care provider, with no care-seeking more common for diarrhea [32].

The national estimates for the use of ORS/RHS sourced from any venue indicate a marginal increase from 41% to 45% between 2008 and 2018. Previous studies in different parts of the country reported 43.5% prevalence of ORS usage in Cross River State [33]; 49.5% in a Military Barrack in Ibadan [34], while Egbewale and colleagues [5] report 39.7% usage across Nigeria. A significant increase from 38% to 55% pre- and post-intervention in the eight Nigerian States covered by the CHAI's program was reported by Schroder and colleagues [35]. Similar to our findings on antibiotic use, there are reported reduction in usage for diarrhea treatment in other developing countries [36, 37].

The positive change in ORS/RHF usage in Katsina, Kano and Bauchi States might be explained by the impact of the ECMPS program undertaken by CHAI [13] but this was not the case for the other states covered by the program; of particular concern is Kaduna where the estimates indicate a negative change that was significant. Studies by Omole and colleagues [38] reported that awareness about ORS for management of diarrhea was universal in Samaru, Kaduna State, but this does not translate to usage as only 34% resorted to using it for their children. An earlier study found widespread use of medicinal plants for treating diarrhea in the State [39]. The dramatically high usage in Borno, a state affected by conflict since around 2009, could be attributed to focused interventions from the international community. The WHO trained a number of community resource persons on integrated community case management of childhood illnesses who were deployed to internally displaced persons (IDP) camps to treat children with mild ailments such as diarrhea, malaria and pneumonia while those with serious conditions were to be referred to nearby health facilities [40]. There were also WHO-supported mobile health teams, a group of trained medical professionals who provide essential health services to vulnerable and displaced populations and host communities across states in the north east region [41].

For the majority of the States, the proportion of children with diarrhea for whom treatment was sought at a health facility was not significantly higher than the national average, although there is evidence that the use of ORS/RHF in Nigeria is shaped by the relative propensities of patients in different areas to seek care at health facilities. Consequently, a community-based approach as practiced in the conflict State of Borno can be considered in other parts of the country.

The study has some limitations. Responses concerning treatment behavior were self-reported by the caregivers and so could be subjected to recall error. The DHS questionnaire that was used to elicit information conatined questions that required the women to indicate the type of care received by the child and if antibiotic pill, syrup or injection was administered. The responses would thus be subject to their ability to correctly identify what was given to the child considering that it is possible for the attending physician to administer a drug without the mother knowing exactly what the drug is.

## Conclusion

We have characterised spatial and temporal trends in the treatment seeking and treatment outcomes for episodes of early childhood diarrhea in Nigeria over the period 2008–2018 covered by three rounds of DHS surveys. While there have been increases in the fraction of cases receiving treatment with recommended oral rehydration therapy at the national level, there has been substantial sub-national variation in these improvements, and antibiotics remain over-used in the treatment of diarrhea across Nigeria. While care seeking at a health facility is associated with an increased likelihood of receiving the recommended oral rehydration therapy, it is also associated with an increase likelihood of antibiotic treatment at a level equal to that of care seeking at a chemist/pharmacy.

## Acknowledgments

Comments received on the work from members of the National Malaria Elimination Programme (NMEP) of Nigeria are appreciated.

## Author Contributions

**Conceptualization:** Ewan Cameron, Peter Gething.

**Data curation:** Ezra Gayawan.

**Formal analysis:** Ezra Gayawan, Ewan Cameron, Osafu Augustine Egbon.

**Funding acquisition:** Peter Gething.

**Investigation:** Ezra Gayawan, Ewan Cameron.

**Methodology:** Ezra Gayawan, Ewan Cameron.

**Project administration:** Tolu Okitika.

**Software:** Ezra Gayawan.

**Supervision:** Peter Gething.

**Visualization:** Ezra Gayawan, Ewan Cameron.

**Writing – original draft:** Ezra Gayawan, Ewan Cameron.

**Writing – review & editing:** Ezra Gayawan, Ewan Cameron, Tolu Okitika.

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
