## [Decision Letter · Decision Letter 0]

20 Feb 2024

PONE-D-23-41643A situational assessment of treatments received for childhood diarrhea in the Federal Republic of NigeriaPLOS ONE

Dear Dr. Gayawan,

Thank you for submitting your manuscript to PLOS ONE. After careful consideration, we feel that it has merit but does not fully meet PLOS ONE’s publication criteria as it currently stands. Therefore, we invite you to submit a revised version of the manuscript that addresses the points raised during the review process.

The study is highly relevant and will guide funders and decision makers in setting formulating priorities in prevention, care and treatment of under-5 diarrhea. Please review and address the comments listed below in addition to reviewers' (1 and 2) comments. Introduction:Consider making this section more concise focusing on the key objectives of the study.

Methods

Please explicitly indicate the study design. Even though DHS data was used, but it will be important for readers to understand the study design.Datasets: the language used sounds like the data was primarily collected for this study. Please adjust the language.Will be good to briefly provide context about Nigeria geopolitical structure and also socioeconomic context of the 6 geopolitical zones (consider a map). This is important because some of the findings are closely related to these parameters.

Results & Discussion:

Please consider splitting this section into “results” and “discussion” sections. The results can be concisely outlined for example in a table under relevant subsections or in any format the authors feel appropriate.Discuss findings chronologically in the discussion section. Subsection “further discussion”: This is confusing. Inline with the structure of the study, results and discussion, consider the following:  moving components in this subsection to relevant subsections; creating a subsection “change in trends” for most of the contents in the “further discussion” subsection.Also ensure that all figures are presented in line with plosone figure guidelines: https://journals.plos.org/plosone/s/figuresPlease submit your revised manuscript by Apr 05 2024 11:59PM. If you will need more time than this to complete your revisions, please reply to this message or contact the journal office at plosone@plos.org. Please include the following items when submitting your revised manuscript:A rebuttal letter that responds to each point raised by the academic editor and reviewer(s). You should upload this letter as a separate file labeled 'Response to Reviewers'.A marked-up copy of your manuscript that highlights changes made to the original version. You should upload this as a separate file labeled 'Revised Manuscript with Track Changes'.An unmarked version of your revised paper without tracked changes. You should upload this as a separate file labeled 'Manuscript'.If applicable, we recommend that you deposit your laboratory protocols in protocols.io to enhance the reproducibility of your results. Protocols.io assigns your protocol its own identifier (DOI) so that it can be cited independently in the future. For instructions see: https://journals.plos.org/plosone/s/submission-guidelines#loc-laboratory-protocols. Additionally, PLOS ONE offers an option for publishing peer-reviewed Lab Protocol articles, which describe protocols hosted on protocols.io. Read more information on sharing protocols at https://plos.org/protocols?utm_medium=editorial-email&utm_source=authorletters&utm_campaign=protocols.

We look forward to receiving your revised manuscript.

Kind regards,

Ibrahim Jahun, MD, MSC, PhD

Academic Editor

PLOS ONE

Journal Requirements:

Gates Foundation [INV-009390 OPP1197730]

Peter Gething was additionally supported through funding provided jointly by Curtin University, the Telethon Trust and the Telethon Kids Institute under project ID RES-61992

Reviewers' comments:

Reviewer's Responses to Questions

**Comments to the Author**

1. Is the manuscript technically sound, and do the data support the conclusions?

Reviewer #1: Partly

Reviewer #2: Yes

2. Has the statistical analysis been performed appropriately and rigorously? 

Reviewer #1: I Don't Know

Reviewer #2: Yes

3. Have the authors made all data underlying the findings in their manuscript fully available?

Reviewer #1: Yes

Reviewer #2: Yes

4. Is the manuscript presented in an intelligible fashion and written in standard English?

Reviewer #1: Yes

Reviewer #2: Yes

5. Review Comments to the Author

Reviewer #1: Review: A situational assessment of treatments received for childhood

diarrhea in the Federal Republic of Nigeria

The study assessed for improvement in the case management of diarrhoea in children in Nigeria based on 3 Demographic health survey rounds conducted between 2008 and 2018. The study stratified changes in the proportion of children receiving oral rehydration therapy by venue of treatment and by state.

The study noted that there was significant improvement in the number of children who were seen in chemist or hospital facilities for diarrhoea who received oral rehydration therapy. However, the total number of children receiving oral rehydration therapy for diarrhoea in children was less than ideal due to state level variation in venue attendance and performance by the sites. The use of antibiotics, not particularly recommended in diarrhoea was rampant with negligible variation across venues. There was welcome reduction in the proportion of children with diarrhoea who received antibiotics in a few states.

Comments

The Study contributes important information which is useful for planning and evaluating interventions in the management of childhood diarrhoea.

The study is acceptable subject to the following major corrections.

1. The abstract section includes the following “Nationally, there have been substantial increases in the proportion of children under 5 years old with diarrhoea receiving the recommended oral rehydration therapy after seeking treatment at either a health clinic (0.57 [0.44{0.69; 95% CrI] in 2008; 0.70 [0.54{0.83] in 2018) or chemist/pharmacy (0.28 [0.17{0.42] in 2008; 0.48[0.31{0.64] in 2018).”

This was referred to in the results section lines 151-155, but the details of the change e.g 2008 values and the 2018 values should be included in the results and discussion section

2. Methods section: More comments on how the information on the history of antibiotic use for childhood diarrhoea was ascertained. How did mothers identify antibiotic pills or syrups- did they show the samples of what they had used, or show the child's prescription, was it simply based on recall, or was it assumed that if the child used any additional medication this was likely to be an antibiotic. More light should be thrown on this and some comments on the justification for and the limitations on the ascertainment of antibiotic use can be included in the discussion.

3. Conclusion: Lines 256-258 “While there have been substantial increases in the fraction of cases receiving treatment with recommended oral rehydration therapy at the national level” Line 148 Result section- says “results shows marginal increase (i.e 5%)”,… and does not to support the above conclusion.

Minor Corrections

1. Introduction:

The section is generally well written,

Line 38. “ Because it is rarely microbial caused….” microbial has been deleted and sentence replaced with ‘Because it is rarely caused by bacteria….” As most cases of diarhoea in children are indeed microbial, just that they are caused by viruses and not bacteria.

2. Typographical corrections e.g spelling of Rehydration- Line 145

3. Respective national values to be included as footnotes in relevant figures.

4. Other corrections as in the attached manuscript

Reviewer #2: This is a very well written manuscript. The technical aspects are very sound, and it makes for easy reading. It does not appear to have any ethical concerns. There are a few typographical/spelling errors but not of any significant magnitude.

6. PLOS authors have the option to publish the peer review history of their article (what does this mean?). If published, this will include your full peer review and any attached files.

Reviewer #1: No

Reviewer #2: **Yes: **Patience Ngozi Obiagwu

---

## [Decision Letter · Decision Letter 1]

6 May 2024

A situational assessment of treatments received for childhood diarrhea in the Federal Republic of Nigeria

PONE-D-23-41643R1

Dear Dr. Gayawan,

We’re pleased to inform you that your manuscript has been judged scientifically suitable for publication and will be formally accepted for publication once it meets all outstanding technical requirements.

Kind regards,

Ibrahim Jahun, MD, MSC, PhD

Academic Editor

PLOS ONE

Additional Editor Comments (optional):

Reviewers' comments:

Reviewer's Responses to Questions

**Comments to the Author**

1. If the authors have adequately addressed your comments raised in a previous round of review and you feel that this manuscript is now acceptable for publication, you may indicate that here to bypass the “Comments to the Author” section, enter your conflict of interest statement in the “Confidential to Editor” section, and submit your "Accept" recommendation.

Reviewer #3: All comments have been addressed

2. Is the manuscript technically sound, and do the data support the conclusions?

Reviewer #3: Yes

3. Has the statistical analysis been performed appropriately and rigorously? 

Reviewer #3: Yes

4. Have the authors made all data underlying the findings in their manuscript fully available?

Reviewer #3: Yes

5. Is the manuscript presented in an intelligible fashion and written in standard English?

Reviewer #3: Yes

6. Review Comments to the Author

Reviewer #3: We are unable to get reviewer 1 to validate the edits. Additionally, after several attempts we were not able to get additional reviewer to validate responses to reviewer 1 comments. To facilitate the manuscript forward I (Academic Editor) have acted as a reviewer to validate whether reviewer's 1 comments have been addressed. I am satisfied with the level of revision done and I can confirm that reviewer's 1 comments have been fully addressed.

7. PLOS authors have the option to publish the peer review history of their article (what does this mean?). If published, this will include your full peer review and any attached files.

Reviewer #3: **Yes: **Jahun Ibrahim

---

## [Editor Report · Acceptance letter]

11 May 2024

PONE-D-23-41643R1 

PLOS ONE

Dear Dr. Gayawan, 

I'm pleased to inform you that your manuscript has been deemed suitable for publication in PLOS ONE. Congratulations! Your manuscript is now being handed over to our production team.

Kind regards, 

on behalf of

Dr. Ibrahim Jahun 

Academic Editor

PLOS ONE